

# Seasonal fluctuations of litter and soil Collembola and their drivers in rainforest and plantation systems

Winda Ika Susanti[1], Valentyna Krashevska[1], Rahayu Widyastuti[2], Christian Stiegler[3], Dodo Gunawan[4], Stefan Scheu[1,5] and Anton M. Potapov[1,6]

[1] J.F. Blumenbach Institute of Zoology and Anthropology, Georg-August Universität Göttingen, Göttingen, Germany
[2] Department of Soil Sciences and Land Resources, Bogor Institute of Agriculture, Bogor, Indonesia
[3] Bioclimatology, Georg-August Universität Göttingen, Göttingen, Germany
[4] Center for Climate Change Information, Agency for Meteorology Climatology and Geophysics, Jakarta, Indonesia
[5] Centre of Biodiversity and Sustainable Land Use, Göttingen, Germany
[6] German Centre for Integrative Biodiversity Research (iDiv) Halle-Jena-Leipzig, Institute of Biology, University of Leipzig, Leipzig, Germany

Corresponding author
Winda Ika Susanti,
winda.ika-susanti@biologie.
uni-goettingen.de

## ABSTRACT

Rainforest conversion and expansion of plantations in tropical regions change local microclimate and are associated with biodiversity decline. Tropical soils are a hotspot of animal biodiversity and may sensitively respond to microclimate changes, but these responses remain unexplored. To address this knowledge gap, here we investigated seasonal fluctuations in density and community composition of Collembola, a dominant group of soil invertebrates, in rainforest, and in rubber and oil palm plantations in Jambi province (Sumatra, Indonesia). Across land-use systems, the density of Collembola in the litter was at a maximum at the beginning of the wet season, whereas in soil it generally varied little. The community composition of Collembola changed with season and the differences between land-use systems were most pronounced at the beginning of the dry season. Water content, pH, fungal and bacterial biomarkers, C/N ratio and root biomass were identified as factors related to seasonal variations in species composition of Collembola across different land-use systems. We conclude that (1) conversion of rainforest into plantation systems aggravates detrimental effects of low moisture during the dry season on soil invertebrate communities; (2) Collembola communities are driven by common environmental factors across land-use systems, with water content, pH and food availability being most important; (3) Collembola in litter are more sensitive to climatic variations than those in soil. Overall, the results document the sensitivity of tropical soil invertebrate communities to seasonal climatic variations, which intensifies the effects of the conversion of rainforest into plantation systems on soil biodiversity.

## INTRODUCTION

Tropical rainforests are shrinking worldwide mainly due to the conversion into agricultural plantation systems. In Indonesia, rainforest conversion into agricultural plantations increased greatly in the last 30 years and this is predicted to continue (*Koh & Ghazoul, 2010*; *Gilbert, 2012*; *Gatto, Wollni & Qaim, 2015*). The conversion of tropical rainforest into plantations is associated with the degradation and destruction of habitats resulting in the loss of plant and animal biodiversity above the ground (*Fitzherbert et al., 2008*; *Clough et al., 2016*). Increasingly, this is also documented for belowground taxa including Collembola (*Sousa et al., 2006*), nematodes (*Krashevska et al., 2019*), testate amoebae (*Krashevska et al., 2016*), spiders (*Potapov et al., 2020*) and litter macro-invertebrates (*Barnes et al., 2014*). Conversion of forest into other land-use systems, especially into monoculture plantations, is also associated with increased seasonal variation in microclimate, particularly air temperature, relative humidity, vapor pressure deficit and soil temperature, being driven in large by canopy openness (*Meijide et al., 2018*). Thus, rainforests are cooler and have more stable conditions than *e.g.*, monoculture plantations of oil palm and rubber, while the latter are drier and have higher vapor pressure deficit especially during the dry season. Additionally, forests also have lower temperature and moisture amplitudes (maximum-minimum) compared to monoculture plantations. However, the temporal changes of tropical soil animal communities and the influence of microclimate on these changes are little explored.

Investigating responses of belowground invertebrates to land use and (micro)climatic changes are crucial to understand the functioning of tropical ecosystems because belowground biodiversity sustain decomposition, nutrient cycling and water infiltration (*Petersen & Henning, 1982*; *Bardgett & van der Putten, 2014*). Moreover, responses of belowground organisms to land use may be different from those aboveground. For example, in temperate grasslands aboveground diversity is promoted by diverse surrounding land-cover, while belowground diversity is positively related to a high permanent forest cover in the surrounding landscape (*Provost et al., 2021*). *Krashevska et al. (2022)* revealed that land-use change shifts and magnifies seasonal variations of the belowground ecosystem, especially the structure and functioning of microbial communities. Among soil animals, Collembola is a dominant group, abundant and diverse in tropical rainforests and plantation systems (*Hopkin, 1997*; *Devi, Singh & Devi, 2011*; *Susanti et al., 2021*; *Potapov et al., 2021*). They significantly affect soil microbial communities, nutrient cycling and soil fertility by feeding on soil microorganisms and dead organic matter (*Rusek, 1998*; *Coulibaly et al., 2019*). Density and community composition of Collembola were shown to be affected by seasonal variations in microclimate in temperate forests (*Hutson & Veitch, 1987*; *Xu et al., 2012*). For example, *Mayvan, Shayanmehr & Scheu (2015)* reported Collembola density in seasonal temperate forests (Iran) to be the lowest in the dry season. In the tropics, where seasonal climatic variations are less pronounced, seasonal fluctuations in Collembola density and community composition have been reported in few studies, with also usually lower density in the dry than in the wet season (*Palacios-Vargas & Castaño-Menesesm, 2003*;

*Wiwatwitaya & Takeda, 2005*; *Muturi et al., 2009*). *Krashevska et al. (2022)* also reported that soil microfauna, such as Collembola and Mesostigmata are less affected by seasonal variation as compared to soil macrofauna, such as Coleoptera, Psocoptera, and Diptera. However, little is known on changes in seasonal variations in Collembola communities in land-use change context, *i.e.*, due to the conversion of forest into agricultural plantation systems, particularly in the tropics.

Community composition of litter and soil Collembola is linked to both biotic and abiotic factors. *Petersen (2011)* reported Collembola community composition in temperate shrublands to depend on temperature and water content with these factors likely affecting Collembola directly. Functional groups and species of Collembola were found to depend mainly on local abiotic factors such as pH and soil water content in forests in both temperate (*Salamon, Scheun & Schaefer, 2008*) and tropical ecosystems (*Susanti et al., 2021*). However, these effects likely are mediated by changes in microorganisms as a key food resource of Collembola (*Filser et al., 2002*; *Pollierer & Scheu, 2017*). Since changes in land use affect both biotic and abiotic factors, this likely leads to changes in seasonal variations in Collembola abundance and community composition. Combined effects of these factors are poorly known.

Seasonal variations in Collembola communities are linked to their vertical distribution in soil and thereby to Collembola ecological groups, so-called 'life forms'. Collembola community composition varies between soil layers and this is closely linked to changes in the relative abundance of Collembola of different life forms (*Salmon et al., 2014*). In turn, life form composition of Collembola communities is linked to their effects on ecosystem functioning (*Potapov et al., 2016a*; *Coulibaly et al., 2019*). There are four main life forms of Collembola related to their distribution across the soil profile: atmobiotic (aboveground-adapted), epedaphic (upper litter-adapted), hemiedaphic (lower litter-adapted) and euedaphic (soil-adapted; *Rusek, 2007*). Euedaphic species have been reported to be less sensitive to environmental and seasonal variations than hemi- and epedaphic species, *i.e.*, those living in the litter and on its surface (*Chauvat, Zaitsev & Wolters, 2003*; *Bokhorst et al., 2012*). Whether the sensitivity of different life forms of Collembola to seasonal variations also varies in tropical ecosystems, however, remains unknown.

To address multiple knowledge gaps mentioned above, here we investigated seasonal fluctuations in Collembola density and community composition in litter and soil in rainforest, rubber and oil palm plantations in Jambi province, Sumatra, Indonesia. Jambi province represents a model region to investigate the effects of rainforest conversion on biodiversity and ecosystem functioning at local and regional scales (*Clough et al., 2016*; *Drescher et al., 2016*), which has been mainly converted into monocultural oil palm (16% of total area) and rubber plantations (12%) (*Gatto, Wollni & Qaim, 2015*). We assessed Collembola communities at four sampling date points in 2017 varying in temperature and humidity: end of the wet season (March), beginning of the dry season (June), end of the dry season (August) and beginning of the wet season (November). We hypothesized that (1) the density of Collembola changes with season, being lowest in the dry season across land-use systems (especially in litter in plantations), (2) seasonal changes in density are less pronounced in soil-adapted (euedaphic) than in surface-adapted (epedaphic and

atmobiotic) species, (3) differences in the community composition of Collembola between rainforest and plantation systems are the most pronounced in the dry season (because plantations may magnify detrimental seasonal conditions), and (4) seasonal variations in the community composition of Collembola are mainly related to pH and water content, as well as to food resources, *i.e.*, microbial community composition.

## MATERIALS AND METHODS

### Study sites

Portions of this text was previously published as part of the thesis (*Susanti, 2022*). The study was conducted in the framework of the EFForTS project investigating in a comprehensive way ecological and socioeconomic changes associated with the transformation of lowland rainforest into plantation systems rubber (*Hevea brasiliensis*) and oil palm (*Elaeis guineensis*) plantations) (*Drescher et al., 2016*). The study sites were located at a similar altitude of 50 to 100 m a.s.l. in area about 80 km in diameter, with the distance between adjacent sites varying between 0.5 and 5 km) (*Drescher et al., 2016*). Soil and litter samples were taken in three land-use systems located in Jambi province, southwest Sumatra, Indonesia. Each land-use system was replicated four times resulting in a total of 12 sites (3 land-use systems × 4 replicates). The climate is tropical and humid with a rainy season from October to April and a dry season from June to September (*Drescher et al., 2016*).

Rainforest represent baseline conditions allowing to evaluate changes due to the conversion into agricultural plantations. Rubber and oil palm plantations were intensively managed monocultures of 13 to 23 and 15 to 22 years, respectively (*Drescher et al., 2016*). Oil palm plantations were established after clearing and burning of jungle rubber, whereas rubber plantations were established after logging of rainforest (*Allen et al., 2015*). Soils in the study region mainly comprise loamy Acrisols with low fertility (*Allen et al., 2015*; *Kotowska et al., 2015*). Oil palm plantations were fertilized twice a year with NPK complete fertilizer (*i.e.*, Phonska and Mahkota), potassium chloride (KCl) and urea (CO(NH$_2$)$_2$) (*Allen et al., 2015*). Manual and chemical weeding (glyphosate and paraquat) took place throughout the year in both rubber and oil palm plantations (*Allen et al., 2015*; *Kotowska et al., 2015*; *Clough et al., 2016*).

### Climatic data

Background climate data for 2017 were obtained from records taken in the framework of EFForTs (Table 1; *Meijide et al., 2018*). In addition, monthly averages of temperature and precipitation in Jambi Province airport over a period of 30 years (1991–2020) are given in Appendix Fig. 1A.

### Soil sampling and analysis

Soil samples were taken in 2017 at four different sampling dates: end of the wet season (March), beginning of the dry season (June), end of the dry season (August) and beginning of the wet season (November). Samples were taken within 50 m × 50 m plots established at each study site (see *Drescher et al., 2016*) with minimum distance of 500 m, but usually

**Table 1 Temperature and moisture in air and soil in rainforest, rubber plantations and oil palm plantations at four sampling dates in 2017 (March, June, August, November).**

|  | Sampling date | Rainforest | Rubber | Oil palm |
|---|---|---|---|---|
| Air relative humidity (%) | March | 98.4 [91.1–100] | 92.2 [72.3–100] | 91.2 [73.1–99.0] |
|  | June | 97.7 [87.7–100] | 93.2 [72.9–100] | 91.5 [71.3–100] |
|  | August | 97.5 [88.4–100] | 83.2 [68.0–89.3] | 88.7 [66.5–99.5] |
|  | November | 98.5 [90.8–100] | 93.4 [74.9–100] | 91.5 [69.8–100] |
| Air temperature (°C) | March | 24.0 [22.0–26.9] | 26.0 [22.8–31.5] | 24.6 [21.7–28.6] |
|  | June | 24.3 [22.1–27.2] | 25.6 [22.6–30.5] | 26.0 [23.0–30.6] |
|  | August | 24.1 [22.0–27.0] | 25.4 [23.2–29.1] | 25.8 [22.6–30.7] |
|  | November | 24.2 [22.1–27.2] | 25.5 [22.4–30.3] | 26.1 [22.9–31.2] |
| Soil moisture (30 cm depth) (% of dry weight) | March | 34.3 [33.6–36.2] | N/A | 34.5 [33.0–39.4] |
|  | June | 31.3 [30.7–33.3] | 40.0 [39.3–41.6] | 34.8 [34.2–35.8] |
|  | August | 31.1 [30.5–32.2] | 30.4 [29.9–31] | 30.4 [29.5–32.5] |
|  | November | 34.2 [32.4–39.4] | 33.1 [32.5–34.5] | 33.6 [30.3–40.5] |
| Soil temperature (30 cm depth) (°C) | March | 25.3 [25.1–25.5] | N/A | 27.8 [27.1–28.7] |
|  | June | 25.2 [25.0–25.4] | 26.1 [25.8–26.4] | 27.7 [27.1–28.1] |
|  | August | 25.0 [24.8–25.2] | 25.7 [25.4–26.0] | 27.8 [27.5–28.1] |
|  | November | 25.5 [25.2–25.7] | 25.3 [25.0–25.6] | 28.3 [27.9–28.7] |

**Note:**
Monthly means and range (in brackets) of daily average measurements. N/A – data not available.

more than 1 km between plots. Each sample measured 16 cm × 16 cm taken to a depth of 5 cm of the mineral soil. From each plot at each sampling date, one randomly positioned sample was taken resulting in 48 soil cores in total. Litter and soil were separated in the field and processed separately (96 samples in total). Then, samples were transported to the laboratory for extraction of soil animals. Animals were extracted by heat for 4–7 days until the substrates were completely dry (*Kempson, Lloyd & Gheraldi, 1963*) and stored in 70% ethanol until further processing. Sampling of animals used in this study was based on Collection permit No. 2841/IPH.1/KS.02.04/X/2016 issued by the Indonesian Ministry of Forestry (PHKA), the State Ministry of Research and Technology of Indonesia (RISTEK) and the Indonesian Institute of Sciences (LIPI).

Environmental variables were measured in mixed samples of litter and soil separately, there are five cores per plot within a radius of ca. 2 m around the soil animal sample. There are abiotic factors (pH, water content, C/N ratio) and biotic factors (microbial community composition in litter and soil as indicated by phospholipid fatty acids; *Krashevska et al., 2015*, V. Krashevska, 2015, unpublished data), including the phospholipid fatty acid (PLFA) (relative) markers of Gram-positive bacteria (sum of i15:0, a15:0, i16:0 and i17:0 PLFAs), Gram-negative bacteria (sum of 2OH 12:0, 2OH 14:0, 16:1ω7, cy17:0, 2OH 16:0, cy19:0 and 2OH 10:0 PLFAs), saprotrophic fungi (18:2ω6, 9), algae (20:5ω3), as well as arbuscular mycorrhizal fungi based on the neutral lipid fatty acid (NLFA) 16:1ω5c, and fine root biomass (<2 mm in diameter), see Appendix S1 (litter) and Appendix S2 (soil). Litter and soil pH (CaCl$_2$) was measured using a digital pH meter (Greisinger GPHR

1400A, Regenstauf). Aliquots of litter and soil material were dried at 65 °C for 72 h, milled and analyzed for total C and N concentrations using an elemental analyzer (Carlo Erba, Milan, Italy). Water content (Wet weight, proportion of dry weight), and root biomass in litter and soil were determined gravimetrically. Details on environmental factors are presented in Appendix S1 (litter) and Appendix S2 (soil).

## Species identification

Collembola were sorted into morphological groups under a dissecting microscope (Stemi 508; Zeiss, Jena, Germany) at 50x magnification based on basic morphological characters (body shape, morphology of furca, antennae, number of eyes). Several individuals of each morphological group from each sample were subsequently cleared with Nesbitt solution on a heating plate (50 °C) for 3–10 min. Then, the animals were mounted on slides with Hoyer's solution (for details see *Glime & Wagner, 2017*). Collembola were identified to species level using a compound microscope (Axiovert 35, Zeiss) at maximum 400x magnification using the checklist for Indonesian Collembola (*Suhardjono, Deharveng & Bedos, 2012*) and additional articles containing keys for Collembola of southeast Asia, particularly Indonesia (*Potapov & Starostenko, 2002*; *Potapov, 2012*; *Mateos & Greenslade, 2015*). All identified (morpho)species were deposited in http://ecotaxonomy.org. Below, we refer to all identifications as 'species' for simplicity. Whenever possible, juvenile specimens were ascribed to species by comparing with adults or subadults. To indicate vertical stratification, Collembola were classified into life forms. Four life form groups were distinguished based on *Rusek (2007)* and *Potapov et al. (2016b)*:

- Atmobiotic Collembola - length up to 8–10 mm, brightly and often motley colored, long limbs and full set of ocelli, body shape either round or elongated; mostly inhabiting macrophytes such as grasses, bushes, trunks and branches of trees, but also the litter surface. Examples: *Ascocyrtus cinctus*, *Homidia cingula*, *Isotomurus* cf. *parabalteatus*, *Acrocyrtus* sp.1

- Epedaphic Collembola - strong pigmentation, fully developed furca and appendages, complete set of pigmented eyes (8 + 8), medium or large-sized, pronounced but frequently uniform coloring; typically colonizing the litter layer. Examples: *Folsomides centralis*, *Rambutsinella* cf. *scopae*.

- Hemiedaphic Collembola - reduction of body pigmentation and eye numbers (and/or eyes' pigmentation), poorly developed furca, medium or small-sized, inhabiting partly decomposed litter, upper soil layers, or rotten wood. Examples: *Folsomides parvulus*, *Alloscopus tetracanthus*.

- Euedaphic Collembola - absence of pigmentation and eyes, often also furca reduced, typically elongated soft body of medium or small size; largely inhabiting the upper mineral layers of the soil (humus horizon). Examples: *Pseudosinella* sp.1, *Isotomiella* spp., *Megalothorax minimus*.

## Statistical analysis

In all analyses, we used individual soil samples as replicates (litter and soil separately), *i.e.*, $n = 96$ (4 sampling dates × 3 land-use systems × 4 replicates × 2 layers). Statistical analyses were performed with using R statistical software v4.1.3 (*R Core Team, 2022*). We inspected the effect of land-use system ('System'; rainforest, rubber, oil palm), layer ('Layer'; litter, soil) and sampling date ('Season'; March, June, August, November) as well as their interactions on the abundance of Collembola. Generalized linear models with these factors were run using *glmer.nb* (negative binomial distribution) in the *lme4* package v. 1.1-21 (*Bates et al., 2015*). Soil core was included as random effect to account for interdependency of soil and litter samples of the same soil core. The same analysis was applied for each of the four life forms separately.

To assess differentiation in community composition of Collembola species in different seasons and across systems, we applied linear discriminant analyses (LDA). LDA was used as the method that maximize the variance among groups. Only species occurring in at least three plots were included in the analysis. Non-metric multidimensional scaling (*metaMDS*) with six dimensions and 999 permutations was done to normalize the data distributions before continuing with LDA. LDA was done using the *MASS* package including all six axes from the NMDS. Wilks Lambda and *p*-values were used for inspecting the effect of season and land-use system on community composition were calculated using *manova* in the *pander* package v. 0.6.3 (*Daróczi & Tsegelskyi, 2018*). Pairwise tests between land-use systems were conducted using *HotellingsT2* in the *ICSNP* package v. 1.1-1 (*Nordhausen et al., 2018*). To numerically estimate distances among communities of different land-use systems squared Mahalanobis distances ($MD^2$) between land-use systems were calculated using the *mahal* function in the *HDMD* package v. 1.2 (*McFerrin, 2013*).

To identify environmental factors associated with community composition of Collembola across different land-use systems and sampling dates, we used canonical correspondence analysis (CCA) because the lengths of gradients were 3.6 SD units for litter and 3.5 SD units for soil indicating unimodal species - environment relationship (*Jan Leaps, 2003*). We applied forward-selection CCA as implemented in CANOCO 5.02 (*ter Braak & Smilauer, 2012*). Sampling dates and land-use systems were included as silent variables not affecting the ordination. Only species occurring in at least three plots were included in the analyses (Appendix S3 and S4). The following environmental variables were included in the CCA that were *a priori* assumed to affect Collembola community composition: C/N ratio, pH value ($CaCl_2$), water content, PLFA (relative) markers of Gram-positive bacteria, Gram-negative bacteria, fungi and algae, as well as NLFA arbuscular mycorrhizal fungi marker and root biomass. Monte-Carlo tests (999 permutations) were performed to evaluate the overall model significance, and the significance of environmental variables and individual axes. We used forward selection to identify the most important environmental variables affecting Collembola communities. The forward selection procedure was stopped if a variable reached a level of significance > 0.05.
## RESULTS

### Seasonal variations in temperature and moisture

Relative air humidity and soil moisture showed clear seasonal pattern across land-use systems being lowest in August (the peak of dry season), whereas mean as well as maximum and minimum air and soil temperature varied little (Table 1). The long-term climatic data from 30 years (1991–2020) generally showed similar seasonal patterns in precipitation with minimum values during the dry season until August (Appendix Fig. 1A). However, compared to the long-term average precipitation in 2017 was higher during the wet season from September to December being particularly high in November (Appendix Fig. 1B). Overall, therefore seasonal variations in precipitation were more pronounced in 2017 than the long-term average.

### Seasonal variations in total Collembola density

In total, 9,543 Collembola individuals were assigned to 54 species from 27 genera and 13 families across all land-use systems and four sampling dates. Collembola density varied between land-use systems, layers and seasons with the three-factor interaction being significant (Table 2, Appendix S5). On average, across seasons the density of Collembola in litter was highest in rainforest (8,410 ± 1,388 ind. m$^{-2}$, coefficient variation 16.5%) and lowest in oil palm plantations (1,704 ± 1,435 ind. m$^{-2}$, coefficient variation 84%) (Fig. 1A) as well as in soil the density of Collembola also was highest in rainforest (3,779 ± 2,169 ind. m$^{-2}$, coefficient variation 57.3%) and lowest in oil palm plantations (2,224 ± 749 ind. m$^{-2}$, coefficient variation 33.6%) (Fig. 1B). The total density of Collembola in litter and soil across seasons also represents in Fig. 1C. However, the variation of total density in both layer (soil and litter) was smaller than those in individual layer, coefficient variation of total density in rainforest, rubber and oil palm plantation are 21.1%, 24.9% and 21.2%, respectively. In rainforest litter, Collembola density was highest in the wet season (November, March) (Fig. 1A). However, at the end of the wet season (March) changes in Collembola density in litter contrasted between rainforest and oil palm plantations. The density of Collembola in litter of rainforest was lowest at the beginning of the dry season (June), whereas in oil palm plantations, it was lowest at the end of the dry season (August). By contrast, in soil the density of Collembola in each of the land-use systems was highest at the end of the dry season (August) and decreased thereafter at the beginning of the wet season from August to November (Fig. 1B).

### Seasonal variations in density of Collembola life forms

On average across seasons and layers, atmobiotic and euedaphic species dominated in rainforest and rubber, whereas epedaphic species dominated in oil palm. Generally, seasonal variations in all three land-use systems were more similar in soil than in litter. Similar to Collembola in total, in soil the density of each of the Collembola ecological groups typically reached a maximum at the end of the dry season (August), whereas in litter the pattern was less consistent. The density of each of the Collembola life forms was significantly affected by Season, System and Layer, with two or three of these factors interacting (Table 2, Fig. 2). In litter of oil palm, the density of ecological groups of

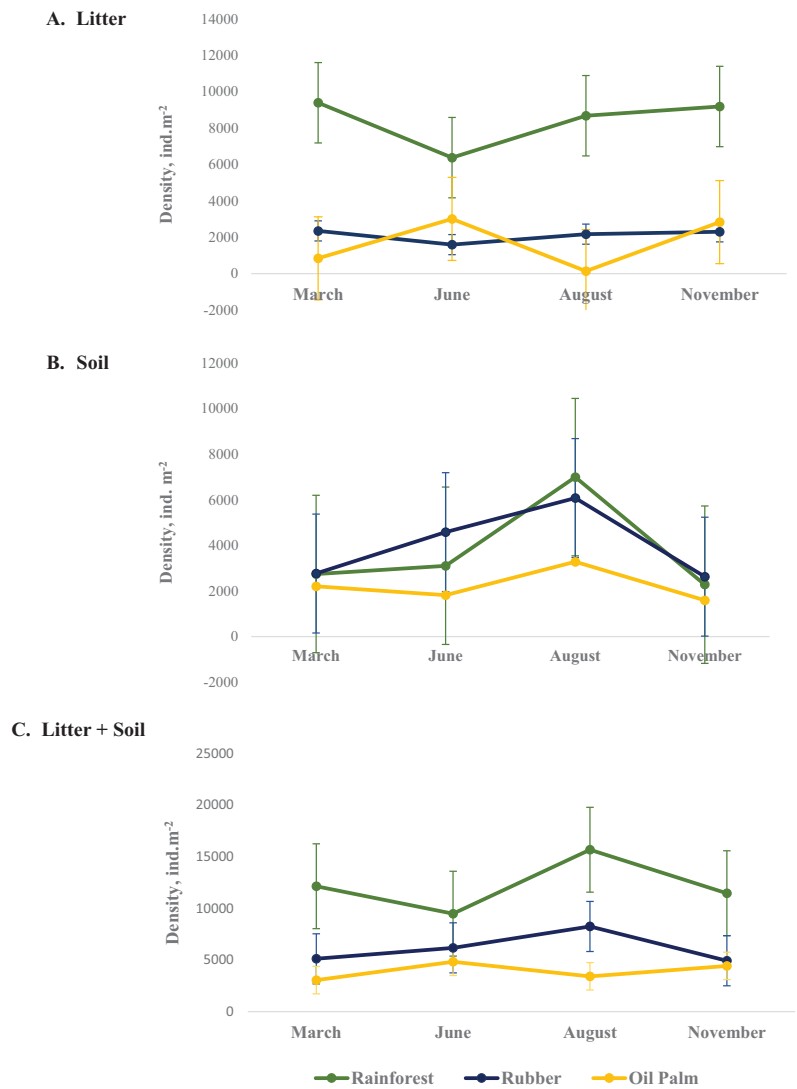

**Figure 1 Density of Collembola in litter (A), soil (B) and litter + soil (C) of different land-use systems (rainforest – green, rubber – blue, oil palm – yellow) at different sampling dates (March, June, August, November).** Error bars represent confidence intervals. Note that when confidence intervals don't overlap, the difference between groups is statistically significant. However, when there is some overlap, the difference might still be significant.

Collembola uniformly was at a minimum at the end of the dry season (August) and reached maximum values in June and November (Fig. 2). By contrast, in litter of rainforest maximum density varied among ecological groups; atmobiotic species peaked in November, epedaphic species in June and August, hemiedaphic species in August and euedaphic species in March. Also, in litter of rubber plantations maximum density varied between Collembola ecological groups but less than in rainforest; atmobiotic species peaked in August, epedaphic species generally varied little with season, and hemiedaphic and euedaphic species peaked in November.

**Table 2 F- and p-values based on linear mixed-effects models on the effect of System (rainforest, rubber and oil palm plantation), Layer (litter, soil) and Season (March, June, August, November) on total density of Collembola, and on the density of atmobiotic, ep.**

|  | Df | F-value | p-value |
|---|---|---|---|
| TOTAL |  |  |  |
| System | 2 | **48.20** | **<0.001***** |
| Layer | 1 | 0.06 | 0.79 |
| Season | 3 | **9.50** | **0.023***** |
| Layer × Season | 2 | **16.10** | **<0.001***** |
| Layer × System | 3 | **11.58** | **0.008**** |
| System × Season | 6 | **19.96** | **0.002**** |
| Layer × System × Season | 6 | **19.61** | **0.003**** |
| ATMOBIOTIC |  |  |  |
| System | 2 | **45.92** | **<0.001***** |
| Layer | 1 | 0.07 | 0.78 |
| Season | 3 | **17.38** | **<0.001***** |
| Layer × Season | 3 | 5.04 | 0.16 |
| Layer × System | 2 | **45.81** | **<0.001***** |
| System × Season | 6 | 7.09 | 0.32 |
| Layer × System × Season | 6 | 8.20 | 0.22 |
| EPEDAPHIC |  |  |  |
| System | 2 | 5.19 | 0.07 |
| Layer | 1 | **17.76** | **<0.001***** |
| Season | 3 | 6.44 | 0.09 |
| Layer × Season | 3 | **21.92** | **<0.001***** |
| Layer × System | 2 | 0.09 | 0.95 |
| System × Season | 6 | **18.42** | **0.005**** |
| Layer × System × Season | 6 | **12.86** | **0.04*** |
| HEMIEDAPHIC |  |  |  |
| System | 2 | 5.20 | 0.07 |
| Layer | 1 | **9.15** | **0.002**** |
| Season | 3 | **1.51** | **0.002**** |
| Layer × Season | 3 | **8.79** | **0.032*** |
| Layer × System | 2 | 2.92 | 0.23 |
| System × Season | 6 | 11.43 | 0.075 |
| Layer × System × Season | 6 | 7.44 | 0.281 |
| EUEDAPHIC |  |  |  |
| System | 2 | **23.35** | **<0.001***** |
| Layer | 1 | 2.39 | 0.12 |
| Season | 3 | 0.99 | 0.80 |
| Layer × Season | 2 | **0.09** | **0.011*** |
| Layer × System | 2 | **11.96** | **0.002**** |
| System × Season | 6 | 7.43 | 0.28 |
| Layer × System × Season | 6 | **18.24** | **0.005**** |

**Notes:**
Df, degrees of freedom; significant effects are given in bold.
* $p < 0.05$.
** $p < 0.01$.
*** $p < 0.001$.

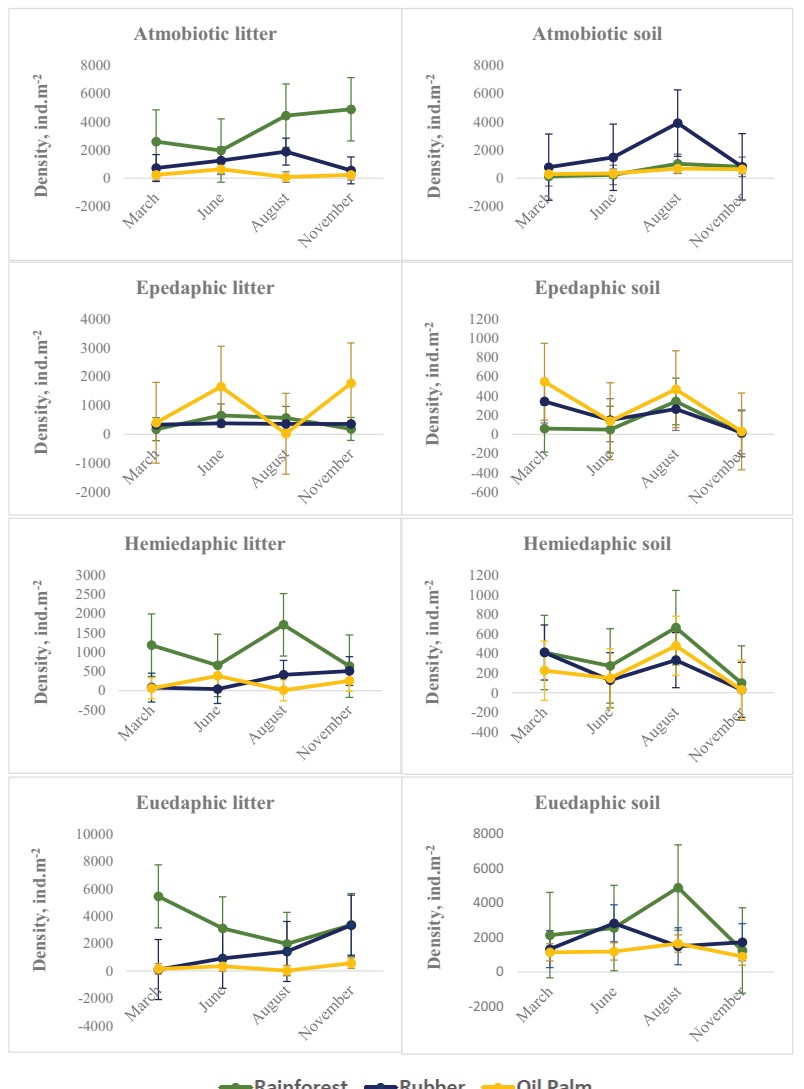

**Figure 2** **Density of Collembola life forms (atmobiotic, epedaphic, hemiedaphic, euedaphic) in litter and soil of rainforest (green), rubber (blue) and oil palm (yellow) at different sampling date (March, June, August, November).** Error bars represent confidence intervals. Note that when confidence intervals don't overlap, the difference between groups is statistically significant. However, when there is some overlap, the difference might still be significant.

## Seasonal variations in the community composition of Collembola

Of the 54 species identified, 54.2% were present in each of the land-use systems, 8.3% were only found in rainforest, 2.1% only in rubber plantations and 6.3% only in oil palm plantations. *Folsomides centralis, Isotomiella* spp. and *Pseudosinella* sp. were most abundant species across all land-use systems. MANOVA based on NMDS scores showed that Collembola community composition varied significantly with Season and System, with the latter depending on Layer (Table 3).

As indicated by LDA, Collembola community compositions differed at every sampling date (Season) between at least two land-use systems with the first LDA axis explaining 65–98% of the variation (Fig. 3; Appendix S6). Across land-use systems and layers the

**Table 3** MANOVA (Pillais trace; based on scores of six NMDS axes) table of (approximate) F- and $p$-values on the effects of Land-use system (rainforest, rubber plantation, oil palm plantation), Layer (litter, soil) and Season (March, June, August, November) on Collem.

| Factor | Df | Pillais trace | F-value | P-value |
|---|---|---|---|---|
| System | 3 | 0.76 | **4.56** | **<0.001**\*\*\* |
| Season | 3 | 0.43 | **2.24** | **0.006**\*\* |
| Layer | 1 | 0.05 | 0.79 | 0.555 |
| Season × System | 9 | 0.85 | **1.57** | **0.013** \* |
| Season × Layer | 3 | 0.30 | 1.52 | 0.098 |
| System × Layer | 1 | 0.25 | **4.45** | **0.001** \*\* |
| Season × System × Layer | 3 | 0.26 | 1.29 | 0.208 |

**Notes:**
Significant effects are given in bold.
\* $p < 0.05$.
\*\* $p < 0.01$.
\*\*\* $p < 0.001$.

separation of communities was most pronounced at the beginning of the dry season (June). In the litter layer of rubber and oil palm plantations Collembola community composition was similar at the end of the rainy season (March). Also, in soil of rubber and oil palm plantations Collembola community composition was similar at the end of the dry season (August). Further, Collembola communities overlapped in soil between rainforest and oil palm plantations at the beginning of the rainy season (November).

## Environmental factors associated with Collembola community composition

In litter, four of the ten environmental variables studied significantly correlated with community composition (CCA, forward selection) explaining 16.7% of the variation in species composition (Trace = 0.57, F = 1.23, $p = 0.004$; Fig. 4A). Water content accounted for 4.9% (pseudo F = 2.2, $p = 0.006$), the fungal PLFA marker 18:2ω6, 9 for 4.3% (pseudo F = 2.0, $p = 0.017$), the arbuscular mycorrhizal NLFA marker 16:1ω5 for 3.8% (pseudo F = 1.8, $p = 0.030$) and litter pH for 3.7% (pseudo F = 1.8, $p = 0.039$). The first axis separated rainforest from rubber and oil palm plantations at all sampling dates, whereas the second axis separated sampling dates in the dry season (June and August) from those in the wet season (November and March) across land-use systems (with the exception of November in oil palm plantations). Certain euedaphic species (*i.e.*, *Isotomiella* cf. *minor*, *Isotomiella symetrimucronata*, *Folsomina onychiurina*, *Thalassaphorura* sp.1, *Megalothorax* cf. *minor*) were associated with high water content in rainforest in the wet season (November, March). A number of atmobiotic and epedaphic species, (*i.e.*, *Dicranocentrus* sp.1, *Ptenothrix* sp.1, *Homidia cingula*, *Isotomurus* cf. *parabalteatus*, *Folsomides centralis*) were associated with low water content and high pH in plantations in the dry season (June, August). Community composition in rainforest in the wet season (November, March) correlated closely with litter water content, whereas community composition in plantation systems generally correlated with litter pH, but during the wet

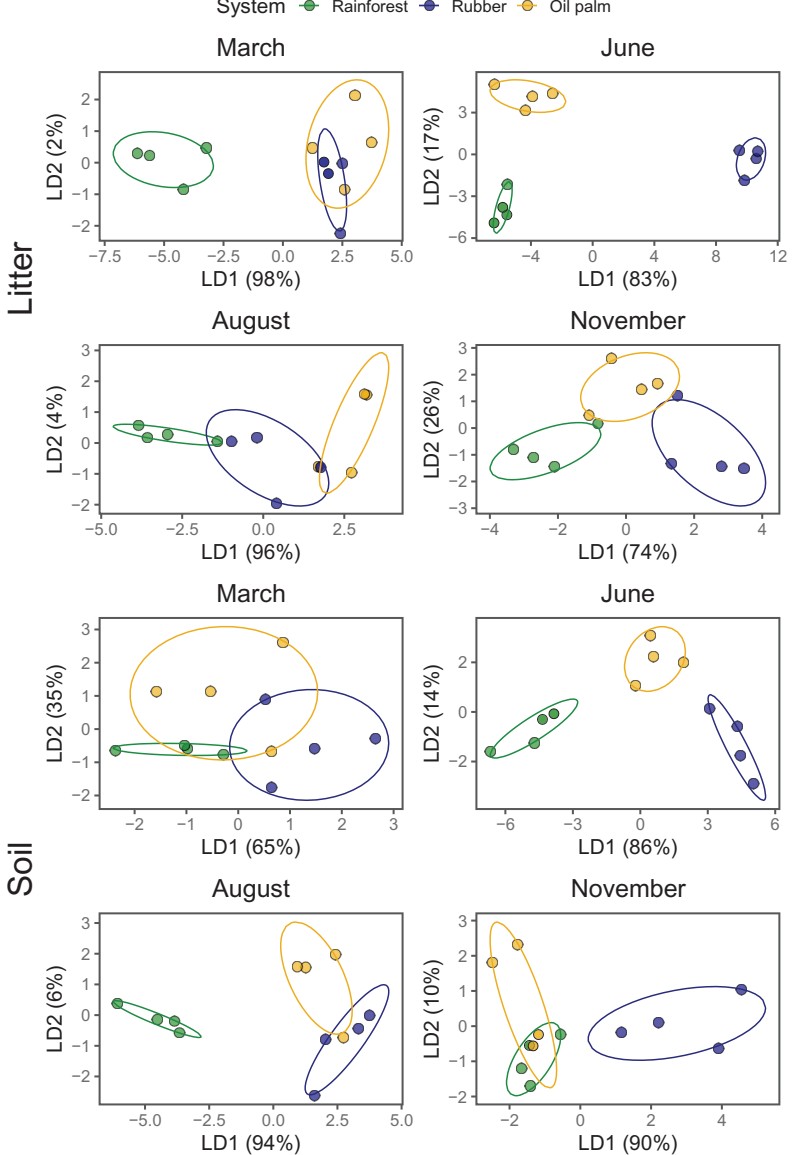

**Figure 3** **Linear discriminant analysis (LDA) of Collembola community composition in litter and soil at different sampling date (March, June, August, November) grouped by land-use systems (rainforest—green, rubber—blue, oil palm—yellow).** Ellipses represent the 95% confidence ranges; each point represents one site.

season (November, March) also with the fungal PLFA and the arbuscular mycorrhizal NLFA marker, especially in rubber plantations.

In soil, environmental factors explained a total of 27.7% of the variation in community composition thereby exceeding that in litter (Trace = 0.54, F = 1.64, $p$ = 0.002; Fig. 4B). Six of the ten environmental variables studied significantly correlated with species composition, with the sum of Gram-positive bacterial PLFA markers accounting for 5.6% (pseudo F = 2.7, $p$ = 0.006), water content for 5.6% (pseudo F = 2.8, $p$ = 0.007), soil pH for 4.5% (pseudo F = 2.4, $p$ = 0.014), C/N ratio for 3.9% (pseudo F = 1.9, $p$ = 0.039), root

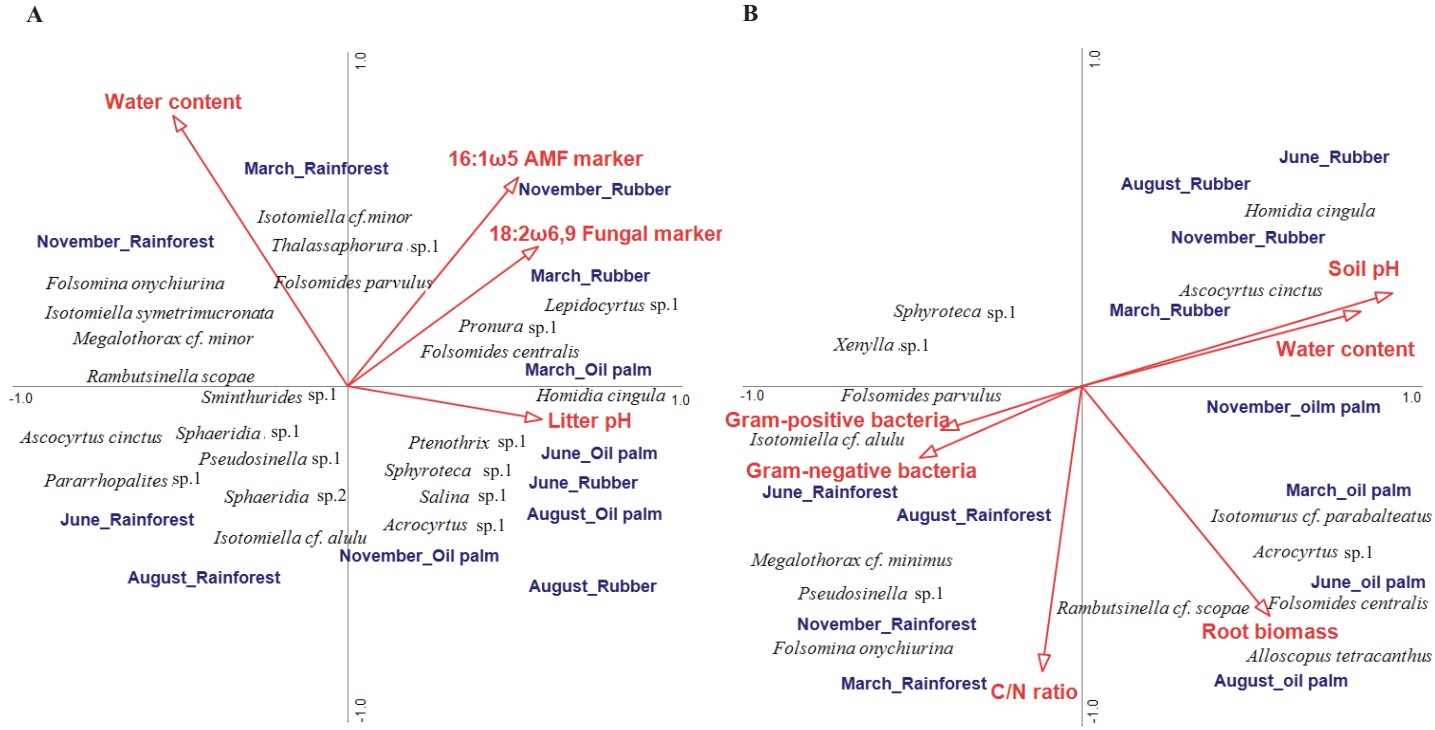

**Figure 4 Canonical correspondence analysis (CCA) of Collembola species in litter (A) and soil (B) as related to environmental variables (red arrows).** Only environmental variables selected as being significant in the forward selection procedure are shown. The three land-use systems (rainforest, rubber plantations, oil palm plantations) and four sampling dates (March, June, August, November) were included as silent variables not affecting the ordination. The length of arrows represents the percentage of variation explained by the environmental variables.

biomass for 4.2% (pseudo F = 1.9, $p$ = 0.044) and the sum of Gram-negative bacterial PLFA markers for 3.9% (pseudo F = 2.0, $p$ = 0.048). Similar to litter, the first axis separated rainforest from rubber and oil palm plantations across sampling dates. Soil pH correlated closely with soil water content and was associated with Collembola community composition in rubber and oil palm plantations across seasons, and in rubber with high density of atmobiotic species such as *Homidia cingula* and *Ascocyrtus cinctus*. Root biomass correlated positively with Collembola community composition in oil palm plantations except for the beginning of the wet season (November) and was associated by high density of *Alloscopus tetracanthus*, *Rambutsinella* cf. *scopae* and *Folsomides centralis*. Further, C/N ratio correlated with Collembola community composition in rainforest in the wet season (November, March) with high density of *Pseudosinella* sp.1, *Folsomina onychiurina* and *Megalothorax* cf. *minimus*. Gram-negative and Gram-positive bacterial PLFA markers were associated with rainforest in the dry season (June, August) and high density of *Folsomides parvulus* and *Isotomiella* cf. *alulu*.

## DISCUSSION

Our study provided first insights into differences in seasonal fluctuations of tropical invertebrate communities between rainforest and plantations. Seasonal climatic changes affected both density and community composition of Collembola across the studied

land-use systems. In general, the density of Collembola was higher in the wet than in the dry season, but the fluctuation pattern was different in soil and litter, as well as in species of different life forms. Differences in density and community composition of Collembola between rainforest and plantations were most pronounced in the litter layer. Water content and pH were identified as the most important abiotic factors associated with Collembola communities, while microbial community composition (Gram-positive and Gram-negative bacteria, saprotrophic and arbuscular mycorrhizal fungi) was identified as an important biotic factor. Overall, Collembola density was higher in rainforest than in rubber and oil palm plantations, which is conform to previous studies on other invertebrate taxa at the same study sites such as ants (*Nazarreta et al., 2020*; *Rizqulloh et al., 2021*) and spiders (*Potapov et al., 2020*; *Junggebauer et al., 2021*).

## Seasonal variations in Collembola density

Partly confirming our first hypothesis, the density of Collembola changed with season being generally higher in the wet than in the dry season. The patterns are consistent with the results of multiple studies showing that higher litter water content in wet season is associated with higher density of Collembola, and generally soil invertebrates (*Muturi et al., 2009*, *2011*; *Detsis, 2000*; *Wiwatwitaya & Takeda, 2005*; *Mayvan, Shayanmehr & Scheu, 2015*). Collembola have physiological adaptation to seasonal variation including timing mechanisms of egg and reproductive stages especially in dry season due to the seasonal temperature regime which has a strong effect on phenology (*Vegter, 1987*). Low soil moisture was reported to be the most important mortality factor for Collembola (*Stamou et al., 1993*).

In litter, the density of Collembola increased in the wet season, particularly early in the wet season (August to November), whereas in soil it strongly decreased, which may have been due to waterlogging and migration into the litter (*Detsis, 2000*; *Krab et al., 2010*). In litter, differences in Collembola density in rainforest and oil palm plantations were more pronounced in the dry season than in the wet season suggesting that water content is an important factor driving Collembola density with detrimental effects of low water content being more pronounced in oil palm plantations. Further, the thick litter layer in rainforest presumably buffers unfavorable conditions for Collembola during the dry season by maintaining higher soil moisture and providing ample food substrate (*Cortet & Poinsot-Balaguer, 1998*). In soil, Collembola density was at the maximum at the end of the dry season (August), suggesting that Collembola thrive throughout the dry season and at least in part move from litter into soil to avoid desiccation. Overall, the results indicate that seasonal changes in water content function as important drivers of the population density and vertical stratification of Collembola communities in tropical ecosystems.

Supporting our second hypothesis, the density of euedaphic species was somewhat less affected by season than that of the other life forms. In particular in rainforest, changes in density of euedaphic species contrasted between litter and soil presumably reflecting that euedaphic species move actively between the layers and suggesting that they may better cope with seasonal climatic changes than other life forms. Supporting this suggestion, euedaphic species *Pseudosinella* sp.1 and *Isotomiella* spp. were the dominant ones at our

study sites (*Susanti et al., 2021*), reflecting that morphological adaptations to life in soil (small and slender body with short appendages and reduced visual apparatus) help to successfully live in tropical soils. Both genera were reported *Pseudosinella* to be well adapted to changes in water content (*Christiansen & Culver, 1969*; *Tranvik & Eijsackers, 1989*; *Adis & Junk, 2002*). High resistance of euedaphic Collembola against harsh environmental conditions is supported by results of previous studies showing that they are able to withstand even extreme weather conditions (*Bokhorst et al., 2012*; *Makkonen et al., 2011*; *Yan et al., 2015*) and may respond little to changes in abiotic factors associated with changes in forest management (*Chauvat, Zaitsev & Wolters, 2003*). Overall, the density and thus the contribution of euedaphic Collembola to ecosystem functioning of tropical soils appears to be more stable in time than that of upper litter-dwelling species.

In soil, seasonal changes of each of the four Collembola life forms were similar irrespective of land-use system with peak densities at the end of the dry season (August) (with the exception of euedaphic species in rubber) and lowest density at the beginning of the wet season (November). By contrast, seasonal fluctuations in litter were generally more variable and differed between land-use systems. In particular in epedaphic species in oil palm seasonal changes in soil were opposite to those in litter – again pointing to the importance of vertical migration for maintaining high population density throughout the year. Generally, the density of epedaphic species in rainforest and rubber was lower than that in oil palm plantations. As indicated by results of our CCA, epedaphic (*Rambutsinella* cf. *scopae* and *Folsomides centralis*) as well as hemiedaphic species (*Alloscopus tetracanthus*) in oil palm plantations benefited from high biomass of fine roots, particularly during dry season (June, August). This suggests that they benefit from root-derived resources (*Li et al., 2021*), which they are probably able to access due to open air-filled pore space in soil during the dry season (*Erktan et al., 2020*).

Atmobiotic species generally reached high density in particular in litter of rainforest and in soil of rubber plantations at the end of the dry season (August). Notably, high density of atmobiotic species at the end of the dry season both in litter and soil suggests that they cope well with dry conditions, which is consistent with earlier studies (*Ponge, 2010*). This is further supported by results of our CCA indicating that particularly at the end of the dry season atmobiotic species (*Ptenothrix* sp.1, *Sphyroteca* sp.1, *Salina* sp.1 and *Acrocyrtus* sp.1) colonize the litter layer of plantations.

An important observation of our study was that vertical stratification of Collembola life forms varied considerably throughout the year. For example, on average the density of atmobiotic Collembola in soil of rubber plantations exceeded that in litter. Vice versa, on average the density of euedaphic Collembola in litter of rainforest exceeded that in soil. This discrepancy between the life form and the actual microhabitat of Collembola suggests that morphological adaptations to certain microhabitats (such as litter and soil) may have little implications for their realized niche in tropical ecosystems with poorly developed soil stratification and relatively shallow litter layer. However, life forms may be informative to predict vulnerability and adaptation of Collembola species to seasonally changing conditions, as we showed above.

## Seasonal variations in Collembola community composition

Confirming our third hypothesis, LDA showed the strongest difference in community composition of Collembola between rainforest and monoculture plantations in the dry season (June). The distinction of community composition between the dry and the wet season both in litter and soil presumably indicates that species in different land-use systems have different adaptations to low water content and high temperature. It is known that the tolerance of Collembola to high temperatures and drought differs between species (*Marx, Guhmann & Decker, 2012*; *Holmstrup, 2019*; *Escribano-Álvarez et al., 2022*; *Alvarez, Frampton & Goulson, 1999*). As water content is a limiting factor for a number of species, the selection force of this factor increases during the dry season, probably explaining strong differentiation between rainforest and plantation communities.

Confirming our last hypothesis, water content and pH were the most important environmental factors associated with the composition of Collembola communities in both litter and soil across seasons. However, again supporting our last hypothesis, we found a number of food-related factors to be associated with Collembola community composition across seasons. Saprotrophic and arbuscular mycorrhizal fungi were associated with Collembola community composition in litter of plantation systems in particular during the wet season. Arbuscular mycorrhizal fungi are abundant in tropical ecosystems (*Soudzilovskaia et al., 2019*) and benefit from high pH in plantations (*Wang et al., 1993*; *Van Aarle, Olsson & Söderström, 2002*). *A'Bear, Boddy & Hefin (2012)* reported arbuscular mycorrhizal fungi to potentially serve as a food resource for springtails under favorable conditions. Dominant genera/species at our study sites, *i.e.*, *Pseudosinella* sp., *Folsomides centralis* and *Isotomiella* spp., have been assumed to predominantly live as fungivores (*Susanti et al., 2021*) and fungivory was suggested to generally dominate in Collembola communities of tropical ecosystems (*Takeda, 1996*). Supporting this assumption, results of our CCA indicated that *Isotomiella* cf. *minor* and *Folsomides centralis* in plantation systems have the same direction with saprotrophic as well as arbuscular mycorrhizal fungi in litter. The correlation between Collembola community composition and fungi in plantations was most pronounced in the wet season suggesting that the role of resources as a driving factor of Collembola community composition in plantations is most pronounced when climatic conditions are beneficial.

In soil of rainforest in the dry season, Collembola community correlated positively with Gram-positive and Gram-negative bacteria. Bacteria is often overlooked as potentially important food resource for Collembola (*Potapov et al., 2021*). Collembola communities in forest ecosystems were shown to correlate with the abundance of bacteria suggesting potential trophic links (*Chauvat, Zaitsev & Wolters, 2003*), which was also supported by our study. Soil C/N ratio was another food-related factor (a proxy for substrate quality) affecting Collembola community composition in soil, in particular in rainforest, in the wet season. This is in line with the results of the study of *Cassagne, Gers & Gauquelin (2003)*, who reported that litter C/N ratio is an important factor driving Collembola community composition in particular at high soil moisture conditions. Microorganisms sensitively react to changes in microclimate and litter quality (*Fierer, Schimel & Holden, 2003*; *Rousk*

& Bååth, 2007; Zhang et al., 2013; Krashevska et al., 2022), changing the available food for Collembola. Some Collembola species inhabiting soil were associated with root biomass indicating that living roots may serve as another factor shaping tropical Collembola communities. Trophic links between plant roots and Collembola were repeatedly shown in forests (Fujii, Saitoh & Takeda, 2014; Potapov et al., 2016a) and agricultural ecosystems, and may reflect both direct feeding on roots and mycorrhizal fungi, but also beneficial effects of root exudates (Li et al., 2021). Overall, our results suggest that in addition to direct effects of microclimatic conditions, Collembola are affected by associated changes in the availability of food resources across seasons. This indirect connection is driven mainly by bacteria and C/N ratio in rainforest (probably representing changing litter decomposition) and mainly by fungi and roots in plantations (probably representing changing root supply). Additionally, management practices in oil palm and rubber plantations, such as weeding, herbicide application and fertilization, impact microclimatic conditions through changes in understory plant cover, soil porosity and water infiltration (Allen et al., 2015). All these changes may contribute to the differences in variations in soil, microbial and animal parameters between rainforest and plantations.

## CONCLUSIONS

Density and community composition of Collembola in rainforest and plantation systems varied significantly with season. In general, Collembola density in the litter layer increased in the wet season (November, March), whereas in soil the density strongly decreased early in the wet season (November) across land-use systems pointing to detrimental effects of waterlogging. Euedaphic species were generally abundant and less affected by seasonal climatic variations. Differences in Collembola community composition in litter and soil were most pronounced during the dry season indicating that land use-induced changes in community composition are increased by drought in plantations. Indeed, water content was identified as a major factor associated with Collembola community composition both in litter and soil across seasons. In addition, factors related to Collembola nutrition indicate that they are influenced by seasonality, with Collembola communities being more structured by fungi and C/N ratios during the wet season and by bacteria and plant roots during the dry season.

Our study is the first to show that the transformation of rainforest into plantation systems not only affects the density and community composition of soil microarthropods, but also their seasonal dynamics. The results document that Collembola sensitively respond to drought with the response being most pronounced in the litter layer of plantations, which needs closer attention in particular in face of global climate change. Management practices targeted at increasing habitat space and food availability in the litter layer and improving soil porosity facilitating vertical movement of Collembola may contribute to strengthening Collembola communities and their functioning in plantation systems. Despite a large sampling effort (almost 10,000 identified individuals), our study represents a single region and a single year and may not be representative for other tropical regions. This emphasizes the need for further studies on seasonal variations of soil animal communities in tropical regions.

# ACKNOWLEDGEMENTS

We thank Indonesian organizations and farmers for granting access to the sampling plots and use of their properties. Special gratitude goes to Feng Zhang, Louis Deharveng and Anne Bedos for verification of Collembola morphospecies identification. We also thank Penelope Greenslade for sharing her unpublished drafts of the key on Indonesian Collembola. This manuscript is the peer-reviewed version of a particular thesis, "Soil fauna in lowland rainforest and agricultural systems of Sumatra: Changes in community composition and trophic structure with focus on Collembola", chapter 5, published by Georg-August Universität Göttingen 2022 (*Susanti, 2022*).

## Funding

This study was funded by the Deutsche Forschungsgemeinschaft (DFG, German Research Foundation)–project number 192626868–SFB 990 in the framework of the collaborative German - Indonesian research project CRC990. Anton Potapov received support from the DFG Emmy Noether program (Projektnummer 493345801) and from the iDiv (DFG–FZT 118, 202548816). The funders had no role in study design, data collection and analysis, decision to publish, or preparation of the manuscript.

## Grant Disclosures

The following grant information was disclosed by the authors:
Deutsche Forschungsgemeinschaft (DFG, German Research Foundation): 192626868.
SFB 990 in the framework of the collaborative German—Indonesian research project: CRC990.
Anton Potapov received support from the DFG Emmy Noether program: Projektnummer 493345801 and iDiv (DFG–FZT 118, 202548816).

## Competing Interests

The authors declare that they have no competing interests.

## Author Contributions

- Winda Ika Susanti performed the experiments, analyzed the data, prepared figures and/or tables, and approved the final draft.
- Valentyna Krashevska performed the experiments, analyzed the data, prepared figures and/or tables, and approved the final draft.
- Rahayu Widyastuti conceived and designed the experiments, prepared figures and/or tables, and approved the final draft.
- Christian Stiegler performed the experiments, analyzed the data, prepared figures and/or tables, and approved the final draft.
- Dodo Gunawan analyzed the data, prepared figures and/or tables, and approved the final draft.

- Stefan Scheu conceived and designed the experiments, authored or reviewed drafts of the article, and approved the final draft.
- Anton M. Potapov conceived and designed the experiments, performed the experiments, analyzed the data, prepared figures and/or tables, authored or reviewed drafts of the article, and approved the final draft.

## Field Study Permissions
The following information was supplied relating to field study approvals (*i.e.*, approving body and any reference numbers):

Sampling of animals used in this study was based on Collection permit issued by the Indonesian Ministry of Forestry (PHKA).

## Data Availability
The raw data about Collembola abundance in soil and litter, the average abundance of Collembola per m2 (per season and land-use), the raw data for environmental parameter measurements, and the climatic data are available in the Supplemental Files.

*Susanti (2022)*. Species matrix of Indonesian litter and soil Collembola with environmental factors [Dataset]. Dryad. https://doi.org/10.5061/dryad.5tb2rbp6b

## Supplemental Information
Supplemental information for this article can be found online at http://dx.doi.org/10.7717/peerj.17125#supplemental-information.

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
