# Peer review of "Seasonal fluctuations of litter and soil Collembola and their drivers in rainforest and plantation systems"

_PeerJ, doi:10.7717/peerj.17125_

## Round 0.1 · original submission · Minor Revisions

Dear authors,

After reading the manuscript and the reviewers' notes, I have opted for the option of minor corrections. I would like to take into consideration the comments of the first and third reviewers, who have read the manuscript carefully and added relevant points to improve it.

Yours sincerely,
Bruno

Reviewer 1 ·

Basic reporting

This article is interesting and useful for broad readers. The article can be published after minor revisions. English is clear. I confirmed raw data in this study. Results seemed appropriate, although it will be better that community abundance among seasonal fluctuation are statistically or numerically compared.

Experimental design

Methods was finely described. They used proper methods to clarify community structure to environmental gradients.

Why did authors use "LDA" instead of "RDA" or "dbRDA" which have been generally used? Please explain the advantage of this statistical method.

Validity of the findings

In general, their discussion and conclusion seems based on their own results. However, some shortcomings need to be cleared.

L228: To evaluate seasonal variation, some quantitative criteria, such as CV, mean square of ANOVA, and so on should be added. Migration between litter and soil may not be real variation in each landuse. Migration is interesting, itself. Therefore, I recommend that seasonal variation in total density (litter+soil) should be added, besides dynamics of density in each layer.

L291-321: Species responses are also interesting. Were the described these species responses in main text were based on statistical analyses? Otherwise, did you visually justify their responses to each variables? It should be useful that the significant correlations between dominant species and some environmental variables should be added to main text or supplementary.

Additional comments

L29-30: "more sensitive" may be better than "less buffered".
L52-53: What and/or how are different between above- and belowground communities in response to landuse?

Reviewer 2 ·

Basic reporting

no comment

Experimental design

no comment

Validity of the findings

no comment

Additional comments

my sugest is change the graphs colors to gray scale and use differents line types to ilustrate the areas sampled.

Reviewer 3 ·

Basic reporting

I found an interesting study because it combine a detailed description of the Collembolan communities (1-year) in soil and in litter, across three environments that were characterized by important environmental and biotic predictors. The diagnosis of the four life form groups is needing some standardization in its description (current version is hard to follow). Although the ‘statistical analysis’ section is short, the manuscript is full of comparisons and models, which turn out to be to reader hard follow the text (stratum, season, ecological group, land-use system). Would be worth to try reduce such comparisons in the main text and keep only the essential.
The Discussion section is well-written and has many important information on the studied system (i.e. , dynamic of the collembolan community along 1-year).

Other comments:

Line 174: medium or large-sized;
Line 180: medium our small-sized;
Line 185: please, cite R accordingly;
Line 192: please, you need provide the reasonable for LDA to assess differences in composition; LDA assumed groups, a priori. So, it is unclear what is being analyzed here. Further, there are another set of compositional analyses, NMDS and CCA, and I have found hard to understand the strategy for composition; if the goal was maximize differences among groups based on compositional axes of NMDS, that should be better explained;
Line 232-233: so, what? Which is the objective of analyze three-way interactions in these models and just to cite that a few were significant?
Line 294: why is it necessary in the Results Section include details of fungal PLFA or arbuscular mycorrhizal NFLA markers? Same for Figure 4;
Lines 296-299: this belongs to community composition (rather than environmental factors associated with).

Experimental design

no comment

Validity of the findings

no comment

Additional comments

no comment

---

## Round 0.2 · accepted · Accept

After the second round of revisions, I am in favor of accepting the manuscript. I just ask you to check the scores of reviewer 3 (minor corrections to the text).
Congratulations!

Reviewer 1 ·

Basic reporting

no comment
I confirmed their revisions.

Experimental design

no comment

Validity of the findings

no comment

Additional comments

I confirmed their all revisions. No more comments to this article.

Reviewer 3 ·

Basic reporting

Thank you for your clear responses to my points on the manuscript. After read the revised manuscript, have just a few other suggestions, most of them related to the reference section, which still needs a careful revision:

- line 172-184: would be important to cite author’s name for cited nominal species (as well as for each first citation of a nominal species);
- line 538: please, check italic in species names (and along all the text to be sure);
- line 543: check underlines in ‘DOI’;
- line 547-548: remove blanck space after ‘moisture’ and ‘soil’, respectivelly;
- line 553: remove quotation mark;
- line 586: missing italic in journal name;
- line 632: Hymenoptera (!);
- line 636-637: is this reference right? (citation form);
- line 675-676: revise italic in journal name; same to line 683, 685, etc;

Experimental design

No any additional comment to provide.

Validity of the findings

No any additional comment to provide.